# Exercise as a therapy for sickle cell-associated musculoskeletal pain among children: Healthcare professionals' perspectives

Britney Pinamang Gyampo, Isaac Mensah Bonsu*

Department of Physiotherapy and Sports Science, Faculty of Allied Health Sciences, Kwame Nkrumah University of Science and Technology, Kumasi, Ghana

* piceabc@yahoo.com

## Abstract

### Introduction

Musculoskeletal (MSK) pain is one of the most substantial and debilitating complications of sickle cell disease (SCD), particularly in pediatric patients. Although exercise is recognized as a potential therapeutic strategy for managing SCD-associated MSK pain, the knowledge and attitudes of healthcare professionals regarding its use remain unclear. This study aimed to explore healthcare professionals' perspectives on the role of exercise in managing MSK pain in children with SCD.

### Method

Face-to-face semi-structured interviews were conducted with nineteen healthcare professionals (pediatric nurses and pediatricians). All interviews were transcribed verbatim; codes were generated and inductively organized into themes.

### Results

Three major themes were identified that described healthcare professionals' perspectives on using exercise as a therapy for MSK pain in children with SCD. These included understanding exercise in managing MSK pain in SCD, barriers to using exercise in managing MSK pain in SCD, and facilitators for implementing an exercise program for MSK pain in SCD.

### Conclusions

Healthcare professionals acknowledge the potential benefits of exercise in managing musculoskeletal (MSK) pain in children with sickle cell disease (SCD); however, overcoming identified barriers and utilizing facilitators is essential for effective implementation.

**Data availability statement:** All relevant data are within the paper.

**Funding:** The author(s) received no specific funding for this work.

**Competing interests:** The authors have declared that no competing interests exist.

## Introduction

Sickle cell disease is an inherited autosomal recessive condition defined by chronic pain episodes caused by vaso-occlusion [1]. Despite the potential benefits, integrating exercise into the management plans of pediatric SCD patients with musculoskeletal pain by healthcare professionals remains a complex and underexplored area.

SCD is characterized by abnormal hemoglobin, leading to the formation of sickle-shaped red blood cells [2]. These misshapen cells can obstruct blood flow, causing frequent and severe pain episodes, particularly in the musculoskeletal system [3]. Among children with SCD, MSK pain is common and impacts their quality of life, limiting mobility, school attendance, and overall well-being [4,5]. Traditionally, MSK pain management in SCD has relied heavily on pharmacological treatments such as analgesics and opioids [6]. However, long-term use of these medications can lead to side effects and complications, driving a need for complementary, non-pharmacological approaches to improve patient outcomes [7,8]. Exercise has gained recognition as a potential therapeutic intervention for managing musculoskeletal pain in various chronic conditions, including SCD [6]. Studies have shown that exercise has the potential to reduce the risk of various musculoskeletal disorders, including bone issues, blood vessel occlusion, and pain [6,9,10]. In children with SCD, exercise can decrease blood viscosity, offering an effective approach to improving overall musculoskeletal health [6].

Pediatricians and pediatric nurses are directly involved in caring for children with SCD, particularly in managing MSK pain and improving the quality of life for affected children [6]. Their perspectives are essential for understanding the feasibility, potential benefits, and challenges of incorporating exercise into SCD management plans. Additionally, their perspectives on using exercise to manage SCD-related musculoskeletal pain play a key role in determining its implementation and success. Therefore, this study aims to explore the views of pediatricians and pediatric nurses on the use of exercise as a therapeutic approach for managing musculoskeletal pain in children with SCD. By understanding their perspectives, we can gain insights into how exercise can be effectively and safely integrated into managing MSK pain in children with sickle cell disease.

## Method

### Study setting

The study was conducted at two healthcare facilities in the Ashanti Region of Ghana from 24 September to 3 December 2024. The region was divided into two geographical zones. To ensure diverse perspectives from healthcare professionals, one hospital was purposively selected from each zone, one representing an urban setting and the other a rural setting. The hospital located in the urban area is a major center for pediatric care and specialized treatments. It serves a large and diverse urban population and is well-equipped with specialized departments, including pediatric sickle cell care. Meanwhile, the hospital in the rural setting offers primary and secondary care services, focusing on essential healthcare for conditions like sickle cell disease.

While it may have fewer resources, it provides an important perspective on how exercise therapy could be adapted to resource-constrained settings. Healthcare professionals' experiences at this hospital highlight the challenges and potential strategies for implementing therapeutic exercise for sickle cell management in rural areas.

### Study design, participants, and data collection procedures

This was a descriptive phenomenological study, and the report was guided by the Consolidated Criteria for Reporting Qualitative Research [11]. Nineteen (19) pediatric nurses and pediatricians who had worked for at least two years were purposively sampled between August and October 2024. All nineteen (19) pediatric nurses and pediatricians who participated in the interviews were directly involved in the treatment of children with SCD. As these hospitals run dedicated SCD clinics weekly, the participants' clinical experience, as reported in Table 1, includes regular and active participation in the care of children with SCD.

After obtaining ethics approval, BPG approached eligible participants, explaining the study's purpose, benefits, risks, and voluntary participation before securing written informed consent. All eligible participants agreed to be part of the study. Meetings were then scheduled based on participants' availability and preferences. Participant characteristics, including years of clinical experience, are detailed in Table 1. In-depth interviews were conducted individually using a semi-structured interview guide (Supplementary file 1) and audio-recorded with participants' consent. Interviews, conducted in English by researcher BPG in private rooms at each hospital, lasted between 47 and 56 minutes. The questions were framed by research objectives, relevant literature, expert feedback, and pre-testing with one pediatric nurse and two pediatricians in a similar setting. Field notes were taken after each interview to capture observations beyond the audio recording.

### Data processing and analysis

Each interview was carefully reviewed and transcribed verbatim after each session. To maintain data accuracy, transcripts were returned to participants for feedback or corrections. Once confirmed, each transcript was read multiple times, organized, and categorized through successive readings. Two researchers (BPG and IMB) independently coded the transcripts to identify emerging themes, which were then discussed regularly by the entire research team during data collection. This collaborative approach helped resolve differences and build consensus on themes, ensuring that key areas requiring further exploration were addressed in subsequent interviews.

The data were analyzed manually using Braun and Clarke's [12] thematic approach. Researchers approached the analysis with an open mind, setting aside preconceptions and assumptions, and read through the transcripts multiple times to fully understand the data. Codes were developed from the interview transcripts and supported by direct participant quotations. Following O'Reilly and Parker's guidelines [13], thematic data saturation was achieved when no new themes emerged from the data collected at each hospital.

Table 1. Demographic information and characteristics of the participants according to their years of clinical experience and sex (n = 19).

| Participants Characteristics | Healthcare professionals | |
| --- | --- | --- |
| | Pediatrician (n = 8) | Pediatric nurse (n = 11) |
| Sex<br>Male<br>Female | <br>2<br>6 | <br>4<br>7 |
| Year of clinical experience<br>2-5<br>6-10<br>11+ | <br>2<br>3<br>3 | <br>4<br>2<br>5 |

Trustworthiness was established following Lincoln and Guba's [14] criteria of credibility, confirmability, dependability, and transferability. To enhance credibility and confirmability, member checking of transcripts and peer review of themes were conducted. Dependability and transferability were reinforced through detailed field notes and thorough documentation of the study processes.

### Ethical consideration

Ethics approval, with reference number CHPRE/AP/867/24, was obtained from the Committee on Human Research Publication and Ethics. Participants were assured of anonymity and the confidentiality of the information they shared.

### Results

The majority of the 19 participants were female (n = 13) with over 11 years of clinical experience (Table 1).

Data analysis identified three (3) major thematic areas that described healthcare professionals' views on exercise as a therapy for MSK pain in children with SCD in the selected hospitals. These were an understanding exercise in managing MSK pain in SCD, barriers to using exercise in managing MSK pain in SCD, and facilitators for implementing an exercise program for MSK pain in SCD. These have been presented below, with relevant quotes from participants.

### Understanding exercise in managing MSK pain in SCD

This theme described participants' perception of exercise benefits in managing MSK pain among children with SCD. The majority of the participants explained the role of exercise in managing MSK in children with SCD.

*"I know exercise can help improve circulation, and reduce the frequency and intensity of pain crises. Again, exercise can enhance better blood flow and prevent blockages in the blood vessels". (Pediatrician, urban hospital)*

*"Exercise helps alleviate their pain. For instance, when they engage in physical activity and their muscles and joints are in motion rather than at rest, it can reduce the pain they're experiencing." (Pediatrician, Urban Hospital)*

*"Children with SCD often experience stiffness and joint pain due to limited mobility. Engaging in exercise can help strengthen their bones and joints, enhance their range of motion, and improve joint flexibility" (Pediatric nurse, rural hospital)*

Participants reported how exercise can manage MSK pain among children with SCD; however, they also emphasized the importance of moderation and avoiding overexertion and unnecessary strain.

*"Exercise supports their mobility, particularly in improving joint movement. For those who struggle to walk, it can be especially beneficial. However, excessive exercise may not be helpful due to the pain they experience." (Pediatric nurse, urban hospital)*

*"I think it enables the free flow of blood at the joints, and they're not to engage in extreme or vigorous exercises, but some passive ROM exercises" (Pediatrician, rural hospital)*

*"The more active children are, the more they can cope with the pain of sickle cell disease. It's not about intense exercise, but staying active helps keep their muscles from getting too tight."(Pediatric nurse, rural hospital)*

Participants mentioned various types of exercises to manage musculoskeletal pain in children with SCD. However, most did not specify the intensity or duration, indicating a limited understanding of appropriate exercise dosages.

*"I know that weight-bearing exercises help children with SCD strengthen their bone density, but I'm not certain about the appropriate dosage." (Pediatrician, rural hospital)*

*"There's something about movement that helps with the pain. Exercise like active walking we've tried seems to help a child with SCD muscles feel looser and less prone to cramping."(Pediatric nurse, rural hospital)*

**Barriers to using exercise in managing MSK pain in SCD**

The study also explored participants' perspectives on the challenges of using exercise to manage MSK pain associated with sickle cell disease. These reported barriers could be categorized into three: pain, lack of family members' support, and resource limitation.

Many participants expressed concern that exercise might worsen existing pain or potentially trigger new episodes of pain.

Indeed, the previous section has presented participants' perceptions of the benefits of exercise in managing MSK pain among children with SCD. Yet, many pediatric nurses and pediatricians in this study identified pain as a major barrier to incorporating exercise into the management of sickle cell-associated MSK pain. They suggested that administering analgesics could help reduce pain, making exercise easier for patients.

*"I think the pain is the main issue. I am wondering how a child in pain will be able to tolerate exercise. When the child is in pain, he/she won't even allow you to touch him/her. However, I believe that we can administer analgesics like morphine, brufen, tramadol, diclofenac, or hydroxyurea before they start exercising to reduce the pain level" (Pediatrician, urban hospital)*

*"It's pain. When there is pain, the children may not engage in any form of physical activity, even walking. When the children are given pain relief medication, whether admitted or discharged, they can engage in some form of activities". (Pediatric nurse, urban hospital)*

*"Mostly when they have crises (Vaso-occlusive crises), they can't do major activities because there's pain". We always give analgesics to reduce pain. Perhaps, they can engage in some form of physical activity after that (Pediatric nurse, rural hospital)*

Others also explained that family members sometimes avoid encouraging exercise out of fear that it might worsen pain or lead to health complications. This protective instinct, though well-intentioned, can limit children's physical activity, which is often essential for managing musculoskeletal pain.

*"I normally perform simple stretching exercises when these children are admitted to the ward. When the child experiences little pain when stretching the joints, the mothers tell me to stop and allow the child to rest for a while. I try explaining the benefits; however, this mindset can become a barrier" (Pediatric nurse, urban hospital)*

*"Parents can sometimes act as barriers to exercise. When simple exercises, like stair-walking, are recommended, they may hesitate to encourage their child due to concerns about worsening the pain. Unless you clearly explain the benefits, some parents might not understand or even firmly refuse to allow the activity" (Pediatrician, urban Hospital).*

Finally, limited resources hindered the use of exercise in managing MSK pain associated with sickle cell disease. Some participants expressed that they face challenges in designing individualized, safe exercise programs that avoid risks since they are not experts in exercise prescription. They mentioned that access to physiotherapists and exercise professionals experienced with chronic illness management is limited. This lack of trained support has led them to recommend only the exercises they are familiar with.

*"I am familiar with basic exercises like walking and stretching that can help the children improve strength and flexibility. I usually recommend these when patients are admitted to the ward. I also advise the mothers to encourage their*

children to do the same after discharge. Although I am not an expert in exercise prescription, I do so because the closest hospital with a physiotherapy department is about a 1.5-hour drive away." (Pediatric nurse, rural hospital)

"I understand the importance of exercise for managing musculoskeletal pain, but without expertise in exercise prescription, I am hesitant to suggest anything beyond basic recommendations. When I refer patients to specialists, such as physiotherapists or exercise therapists, parents often complain about the long distance to the physiotherapy unit."(Pediatrician, rural hospital)

**Facilitators for implementing an exercise program for MSK pain in SCD**

Participants described the main facilitators that make implementing an exercise program to manage musculoskeletal (MSK) pain in children with sickle cell disease (SCD) more feasible and effective. Many participants highlighted healthcare collaboration and family involvement as facilitators, as reflected in the quotes below:

I believe that working with physiotherapists and other specialists will ensure that the exercise program is safe and will meet each child's specific needs, especially given the unique challenges of SCD."(Pediatrician, rural hospital)

Additionally, one participant mentioned that interdisciplinary collaboration will enhance family confidence in implementing the exercise program.

"Parents are particularly happy when they see different healthcare professionals treating their children, especially when they see these professionals around the child's bedside. I am sure when they see the physiotherapist, doctor, nurses, and other professionals discussing the child's condition, it will boost their confidence in any treatment that will be given and encourage better adherence."(Pediatric nurse, urban hospital)

Others suggested that physiotherapists and exercise prescription experts should be present in the consultation rooms on clinic days to offer their expert opinions.

"If physiotherapists could join us in the consultation rooms, it would streamline the process". (Pediatrician, urban hospital)

"It would be so helpful to have a physiotherapist on clinic days. They can assist in the assessment process and provide expert advice on exercises that we might not be able to offer on our own."(Pediatrician, rural hospital)

Family involvement is an important factor in successfully implementing exercise programs for managing MSK pain in children with SCD. Some participants believed that when families are educated about the benefits of exercise for managing MSK pain, they are more likely to support the program implementation.

"When parents, especially the mothers, are educated about the benefits of exercise programs, I am sure they will be more likely to support the child's involvement. Involving parents will make it easier to implement the program."(Pediatric nurse, urban hospital)

"When parents are involved, it means that someone is there at home to keep an eye on how the child is reacting to the exercises. They can assist us in identifying any problems early on, which is essential for making adjustments that are safe and efficient". (Pediatrician, urban hospital)

This section has presented three main thematic areas with supporting statements. In summary, participants were generally familiar with the benefits of exercise in managing MSK pain in children with SCD. However, several barriers and facilitators affecting the implementation of exercise programs to manage MSK pain in SCD were also highlighted.

## Discussion

This study explored healthcare professionals' (pediatricians and pediatric nurses) perspectives on using exercise to manage musculoskeletal pain in children with sickle cell disease. Participants acknowledged several benefits of exercise in managing musculoskeletal pain in children with sickle cell disease. They highlighted advantages such as enhanced circulation, increased flexibility, improved muscle strength, and greater range of motion, all of which can contribute to pain relief and better overall physical function in these children [15]. When participants understand how exercise can alleviate MSK pain in these children, they are better positioned to incorporate it into comprehensive pain management plans [16]. Identifying these benefits allows them to view exercise as a general recommendation and a specific, targeted intervention for effectively addressing MSK pain [16].

Many participants acknowledged that exercises, particularly low-impact exercises like stretching, walking, or range-of-motion activities, are appropriate to reduce MSK pain and discomfort associated with the disease. The fundamental challenge of exercise for patients with sickle-cell disease remains the prevention of exercise-induced vaso-occlusive crises, which can result from dehydration, hyperthermia, and lactic acidosis [17]. However, since moderate-intensity exercise avoids the accumulation of blood lactate, and a facility-based indoor physical activity program allows for proper hydration, controlled temperature, and blood lactate monitoring, a safe and potentially beneficial program of moderate-intensity exercise seems feasible [17]. Such a program, like "play-based & social activities" (indoor recreational games, light ball play, dancing, and performing daily chores under supervision), could offer an effective approach to managing musculoskeletal pain in patients with sickle cell disease [6].

Although healthcare professionals understand the potential benefits of exercise in managing MSK pain among children with SCD, there are knowledge gaps in determining the appropriate dosage, type, intensity, frequency, and duration. Without a clear understanding of safe exercise dosage, there is a risk of either underutilizing exercise as a therapeutic tool or unintentionally prescribing levels of activity that could exacerbate symptoms [6]. The findings indicate a need for enhanced training on the principles of exercise prescription that equips healthcare professionals with the knowledge and confidence to recommend safe, effective exercise for children with SCD. Developing guidelines that clarify safe exercise intensities and types for SCD could support more widespread adoption of exercise as a treatment modality.

This study acknowledges the complex challenges associated with integrating exercise into the management of musculoskeletal (MSK) pain in children with sickle cell disease (SCD). A concern among participants was that vaso-occlusive pain often makes movement and exercise difficult or even intolerable during pain episodes. Healthcare providers emphasized the inherent difficulty of encouraging physical activity in children actively experiencing pain crises, raising important considerations for the practical implementation of exercise interventions in this population.

A key psychological barrier is kinesiophobia, the fear of movement due to pain [18]. In SCD, where pain is unpredictable and severe, this fear is particularly understandable. Children may develop avoidance behaviors, which can be unintentionally reinforced by caregivers and clinicians who prioritize safety and pain prevention [19]. This cycle of fear and inactivity paradoxically worsens long-term outcomes by promoting musculoskeletal deconditioning. To break this cycle, clinicians can adopt several practical strategies. A foundational approach is to structure activity program pacing, co-creating an activity plan with the child that begins with short, manageable sessions of low-impact exercise (e.g., 5–10 minutes of walking) and gradually increases in a patient-directed manner. Using an activity diary can provide visual proof of progress and help desensitize the fear that activity always leads to pain. For more complex cases, a proactive referral to a physiotherapist specializing in pediatric chronic pain can facilitate a tailored, graded exposure program. Meanwhile, a clinical health psychologist can address the underlying challenging thoughts using techniques such as cognitive behavioral therapy.

Despite these challenges, some participants noted that exercise may become more feasible when children are adequately managed with pain-relieving medications, such as nonsteroidal anti-inflammatory drugs or opioids. The use of analgesics can temporarily alleviate pain, thereby enabling more comfortable engagement in physical activity [6]. This

finding highlights the importance of a balanced, multimodal approach to care, where effective pain management strategies are used synergistically with therapeutic exercise to improve physical function and quality of life [20,21].

Importantly, these insights highlight the need to individualize exercise prescriptions, taking into account each child's pain severity, psychological readiness, and response to pharmacologic interventions. Tailored exercise programs that align with the child's fluctuating pain status and functional ability, potentially incorporating graded exposure and education to reduce fear, may help mitigate kinesiophobia and promote sustained physical activity in a safe and supportive manner.

Similar to other studies [5,6], a lack of family support was identified as a barrier to using exercise in managing SCD-associated MSK pain. Participants indicated that perceived families' protective attitudes may prevent the child from exercising. This finding highlights the important role of families, particularly mothers, in supporting the implementation of exercise as a strategy for managing MSK pain in children with SCD. Despite the well-documented benefits of exercise in improving circulation, enhancing joint mobility, and alleviating pain, family perceptions—especially maternal concerns—frequently hinder its adoption [10,22,23]. Families often act as the primary caregivers for children with SCD, placing their perceptions and attitudes at the core of care decisions [24]. However, their overprotectiveness and fear of exacerbating their child's condition frequently result in resistance to incorporating exercise into pain management. Many families associate exercise with increased pain or the risk of triggering vaso-occlusive crises, reflecting a gap in awareness about the benefits of controlled and appropriate exercise interventions [5,6]. Addressing this barrier requires focused educational initiatives. Providing families with evidence-based knowledge about the advantages of exercise, along with practical demonstrations of safe and simple exercises, can help build their confidence and ease their concerns. Actively involving families in the development of care plans can also strengthen their sense of control and increase their willingness to support exercise programs.

Another major challenge was the limited access to physiotherapists and exercise professionals experienced in managing chronic illnesses, including SCD. The shortage of skilled professionals capable of providing evidence-based guidance on exercise further compounded the difficulties faced by participants. This gap in human resources limited the availability of individualized exercise interventions and hindered the establishment of collaborative care models involving multidisciplinary teams [25]. Addressing this issue requires increasing the availability of expertise in exercise prescription tailored to SCD management. Investing in professional training programs for existing therapists and healthcare providers is essential to bridge this gap and ensure the effective integration of exercise into comprehensive care plans.

Participants identified key facilitators that could support the successful implementation of exercise programs to manage musculoskeletal pain in children with sickle cell disease. These facilitators focused on collaboration among healthcare professionals and family involvement, essential for ensuring the program's effectiveness. A coordinated approach involving physiotherapists, pediatricians, nurses, hematologist, exercise specialists, and other professionals is vital for developing and monitoring safe, effective exercise interventions [26]. This collaboration ensures that the programs are evidence-based and tailored to the unique needs of children with SCD. Furthermore, engaging families, especially primary caregivers like mothers, is essential for program success. Family involvement will foster adherence to exercise routines and ensure consistency in the child's rehabilitation efforts [6,24]. Educating families on the benefits of exercise and addressing misconceptions or fears about physical activity in SCD could further strengthen their role as supportive partners in the program.

## Limitations and strengths of the study

The findings are based on data from two hospitals and the perspectives of pediatric nurses and physicians only, which may limit the transferability of the results. Future research should actively include the voices of other critical stakeholders to enrich the findings. For instance, physiotherapists could provide essential expertise on practical exercise prescription and graded movement strategies. Hematologists could offer insights into integrating physical activity with complex medical regimens. Most crucially, including patients with SCD and their families is fundamental to ensuring that any future

exercise interventions are feasible, acceptable, and aligned with their lived experiences and priorities. However, the views from both pediatricians and nurses enriched the findings by incorporating diverse perspectives from two critical healthcare roles. Furthermore, we are the first to report on pediatricians' and pediatric nurses' perspectives on the role of exercise in the management plan for children with SCD-associated MSK pain within the Ghanaian context. Future studies could complement these findings with clinical trials or longitudinal studies to validate the reported benefits and challenges.

## Conclusion

Participants in this study recognized the multiple benefits of exercise in managing musculoskeletal pain in children with SCD. Their understanding reflects an awareness of the advantages of exercise, emphasizing the need for incorporating exercises into the management plans for children with this condition. However, while participants understood the benefits, they also pointed out challenges related to implementing exercise programs, such as pain, lack of family members' support, and resource limitations. Nevertheless, collaboration among healthcare professionals and family involvement were identified as facilitators for implementing an exercise program for MSK pain in SCD. Future studies should include diverse stakeholders and empirical evaluations of exercise programs to validate their effectiveness and address safety concerns.

### Implications for practice

Based on our findings, we propose incorporating physiotherapists or exercise specialists into SCD clinic days to provide real-time assessment, counseling, and the development of individualized activity plans. Again, establish formal channels for pediatricians and nurses to collaborate with hematologists and mental health professionals, ensuring physical activity is addressed as a core component of comprehensive SCD care. Implementing these strategies can help transform exercise from a perceived risk into a standard, managed therapy for improving health outcomes in this population.

## Supporting information

**S1 File. (semi-structured interview guide) examines healthcare professionals' views on exercise for managing musculoskeletal pain in children with sickle cell disease (SCD).**
(DOCX)

## Author contributions

**Conceptualization:** Britney Pinamang Gyampo.

**Data curation:** Britney Pinamang Gyampo.

**Investigation:** Britney Pinamang Gyampo.

**Methodology:** Isaac Mensah Bonsu.

**Resources:** Isaac Mensah Bonsu.

**Supervision:** Isaac Mensah Bonsu.

**Writing – original draft:** Isaac Mensah Bonsu, Britney Pinamang Gyampo.

**Writing – review & editing:** Isaac Mensah Bonsu.

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
