## [Decision Letter · Decision Letter 0]

30 Sep 2025

Dear Dr. Mensah Bonsu,

Thank you for submitting your manuscript to PLOS ONE. After careful consideration, we feel that it has merit but does not fully meet PLOS ONE’s publication criteria as it currently stands. Therefore, we invite you to submit a revised version of the manuscript that addresses the points raised during the review process.

We look forward to receiving your revised manuscript.

Kind regards,

Tomasz W. Kaminski

Academic Editor

PLOS ONE

Journal Requirements:

3. Please include captions for your Supporting Information files at the end of your manuscript, and update any in-text citations to match accordingly. Please see our Supporting Information guidelines for more information: http://journals.plos.org/plosone/s/supporting-information .

Additional Editor Comments:

Dear Authors,

Both reviewers have recommended major revisions, and I agree with their assessment. Please revise the manuscript accordingly and carefully address all reviewer comments in your response.

Best regards,

Tomasz W Kaminski

Reviewer's Responses to Questions

**Comments to the Author**

1. Is the manuscript technically sound, and do the data support the conclusions?

Reviewer #1: Yes

Reviewer #2: Partly

2. Has the statistical analysis been performed appropriately and rigorously?

Reviewer #1: N/A

Reviewer #2: N/A

3. Have the authors made all data underlying the findings in their manuscript fully available?

Reviewer #1: Yes

Reviewer #2: No

4. Is the manuscript presented in an intelligible fashion and written in standard English?

Reviewer #1: Yes

Reviewer #2: Yes

Reviewer #1: This manuscript describes a structured interview study of healthcare providers knowledge and attitudes toward the use of exercise as a therapeutic strategy for managing SCD-associated MSK pain in children with SCD. The healthcare providers were chosen from two healthcare facilities in the Ashanti Region of Ghana. The introduction adequately describes the background literature supporting the study rationale. The study methodology and qualitative analysis are appropriate, and the interview guide is provided as supplemental materials. The discussion adequately addresses the study strengths and limitations. I had a number of questions/suggestions to improve this manuscript:

1. Please provide more detail about the pediatricians and nurses who took part in the interviews. Table 1 provides data on the years of clinical experience, but it is unclear how much of that clinical experience included participating in the care of children with SCD. Further, the context of care is important to the interpretation of these results. I assume these healthcare providers were providing in-patient care for acute vaso-occlusive pain, rather than addressing the management of chronic pain, improving physical activity, or attempting to reduce the frequency of acute pain in an outpatient setting.

2. The introduction and discussion with their associate references also need to be clear about these contexts, as the role of exercise in speeding improvement in the resolution of acute pain may be quite different from its role in improving cardiovascular health, improving physical activity, or moderating the effects of chronic pain.

Reviewer #2: The manuscript addresses an important gap in the literature by exploring healthcare professionals’ perspectives on exercise as a therapeutic option for musculoskeletal pain in children with sickle cell disease (SCD). The qualitative design is well-justified, and the findings highlight both perceived benefits and practical barriers, which will be valuable for clinical practice and future program design.

Strengths:

Original and contextually relevant research (first such study in Ghana).

Strong methodology: purposive sampling, ethical clearance, data saturation, and trustworthiness criteria.

Practical insights into barriers (pain, family attitudes, limited resources) and facilitators (collaboration, family involvement).

Areas for improvement:

Clarity of writing: Some sentences are long and could be streamlined for readability. A thorough language edit would improve flow.

Discussion depth: The link between identified barriers (e.g., kinesiophobia, family concerns) and proposed interventions could be expanded with more practical recommendations.

Limitations: The authors acknowledge the limited scope (two hospitals, only nurses and pediatricians). It would be useful to suggest explicitly how including physiotherapists, hematologists, and families in future studies could enrich findings.

Implications for practice: The conclusion could more clearly emphasize actionable recommendations (e.g., integrating physiotherapists into clinic days, caregiver education programs).

**Do you want your identity to be public for this peer review?** For information about this choice, including consent withdrawal, please see our Privacy Policy

Reviewer #1: No

Reviewer #2: No

---

## [Author Response · Author response to Decision Letter 1]

8 Oct 2025

Response to Reviewer’s comments

Dear Editor,

We genuinely appreciate the reviewer’s valuable feedback on improving the clarity and specificity of the manuscript. We recognize the importance of refining certain aspects to strengthen the overall quality of the study. Please find our detailed responses below for each comment.

Reviewer #1:

Comment: Please provide more details about the pediatricians and nurses who took part in the interviews. Table 1 provides data on the years of clinical experience, but it is unclear how much of that clinical experience included participating in the care of children with SCD.

Response: We thank the reviewer for this important comment. We have now clarified in the manuscript that “All nineteen (19) pediatric nurses and pediatricians who participated in the interviews were directly involved in the treatment of children with SCD. As these hospitals run dedicated SCD clinics weekly, the participants' clinical experience, as reported in Table 1, includes regular and active participation in the care of children with SCD.” Highlighted in RED on page 5.

Comment: Further, the context of care is important to the interpretation of these results. I assume these healthcare providers were providing inpatient care for acute vaso-occlusive pain, rather than addressing the management of chronic pain, improving physical activity, or attempting to reduce the frequency of acute pain in an outpatient setting.

Response: We thank the reviewer. The reviewer's assumption is understandable; however, our study was specifically designed to explore perspectives on exercise, not in the context of acute inpatient pain management, but as a therapeutic strategy for managing persistent musculoskeletal pain and preventing pain crises in the outpatient setting. We intended to investigate a gap in the literature, moving beyond the paradigm of purely pharmacological pain management during crises to explore how pediatricians and pediatric nurses view a proactive, non-pharmacological intervention like exercise for improving long-term health outcomes.

Comment: The introduction and discussion with their associated references also need to be clear about these contexts, as the role of exercise in speeding improvement in the resolution of acute pain may be quite different from its role in improving cardiovascular health, improving physical activity, or moderating the effects of chronic pain.

Response: We thank the reviewer for this critically important observation. We agree with the reviewer that the distinct roles of exercise in an acute pain context versus a chronic health context are fundamentally different. We have addressed these in the manuscript to reflect the chronic musculoskeletal pain and long-term morbidity in children with SCD, separating it from the acute vaso-occlusive crisis model.

Reviewer #2:

Comment: Clarity of writing: Some sentences are long and could be streamlined for readability. A thorough language edit would improve flow.

Response: We thank the reviewer for this feedback. We agree that enhancing the clarity and readability of the manuscript is important. We have performed a thorough language edit throughout the manuscript, focusing on streamlining long sentences and improving overall flow.

Comment: Discussion depth: The link between identified barriers (e.g., kinesiophobia, family concerns) and proposed interventions could be expanded with more practical recommendations.

Response: These have been revised to read as “A key psychological barrier is kinesiophobia, the fear of movement due to pain [18]. In SCD, where pain is unpredictable and severe, this fear is particularly understandable. Children may develop avoidance behaviors, which can be unintentionally reinforced by caregivers and clinicians who prioritize safety and pain prevention [19]. This cycle of fear and inactivity paradoxically worsens long-term outcomes by promoting musculoskeletal deconditioning. To break this cycle, clinicians can adopt several practical strategies. A foundational approach is to structure activity program pacing, co-creating an activity plan with the child that begins with short, manageable sessions of low-impact exercise (e.g., 5-10 minutes of walking) and gradually increases in a patient-directed manner. Using an activity diary can provide visual proof of progress and help desensitize the fear that activity always leads to pain. For more complex cases, a proactive referral to a physiotherapist specializing in pediatric chronic pain can facilitate a tailored, graded exposure program, while a clinical health psychologist can address the underlying challenging thoughts using techniques like cognitive behavioral therapy”. Highlighted in RED on page 15.

Comment: Limitations: The authors acknowledge the limited scope (two hospitals, only nurses and pediatricians). It would be useful to suggest explicitly how including physiotherapists, hematologists, and families in future studies could enrich findings.

Response: We have revised the limitation section to read as “The findings are based on data from two hospitals and the perspectives of pediatric nurses and physicians only, which may limit the transferability of the results. Future research should actively include the voices of other critical stakeholders to enrich the findings. For instance, physiotherapists could provide essential expertise on practical exercise prescription and graded movement strategies. Hematologists could offer insights into integrating physical activity with complex medical regimens. Most crucially, including patients with SCD and their families is fundamental to ensuring that any future exercise interventions are feasible, acceptable, and aligned with their lived experiences and priorities”. Highlighted in RED on page 18.

Comment: Implications for practice: The conclusion could more clearly emphasize actionable recommendations (e.g., integrating physiotherapists into clinic days, caregiver education programs).

Response: The implication for practice has been revised and highlighted in RED on page 19. “Based on our findings, we propose incorporating physiotherapists or exercise specialists into SCD clinic days to provide real-time assessment, counseling, and the development of individualized activity plans. Again, establish formal channels for pediatricians and nurses to collaborate with hematologists and mental health professionals, ensuring physical activity is addressed as a core component of comprehensive SCD care. Implementing these strategies can help transform exercise from a perceived risk into a standard, managed therapy for improving health outcomes in this population.”

---

## [Decision Letter · Decision Letter 1]

5 Nov 2025

Dear Dr. Mensah Bonsu,

Thank you for submitting your manuscript to PLOS ONE. After careful consideration, we feel that it has merit but does not fully meet PLOS ONE’s publication criteria as it currently stands. Therefore, we invite you to submit a revised version of the manuscript that addresses the points raised during the review process.

We look forward to receiving your revised manuscript.

Kind regards,

Tomasz W. Kaminski

Academic Editor

PLOS ONE

Journal Requirements:

**Additional Editor Comments:**

Dear Authors,

I apologize for the longer waiting time during the review process. Reviewer 1 changed their recommendation from major revision to accept after the revisions were submitted. Reviewer 2 was unavailable to re-evaluate the manuscript, but their main concerns were, in my opinion, properly addressed.

To ensure fairness, I invited a third reviewer, who suggested only minor comments. Based on all feedback, the decision has been updated to minor revision.

Thank you for your patience and for the thoughtful improvements made to the manuscript.

Best regards,

Tomasz W. Kaminski

Reviewers' comments:

Reviewer's Responses to Questions

**Comments to the Author**

Reviewer #1: All comments have been addressed

Reviewer #3: (No Response)

2. Is the manuscript technically sound, and do the data support the conclusions?

Reviewer #1: Yes

Reviewer #3: Yes

3. Has the statistical analysis been performed appropriately and rigorously?

Reviewer #1: Yes

Reviewer #3: Yes

4. Have the authors made all data underlying the findings in their manuscript fully available?

Reviewer #1: Yes

Reviewer #3: Yes

5. Is the manuscript presented in an intelligible fashion and written in standard English?

Reviewer #1: Yes

Reviewer #3: Yes

Reviewer #1: (No Response)

Reviewer #3: I really enjoyed reading your manuscript on healthcare professionals’ perspectives on exercise as a therapy for musculoskeletal pain in children with sickle cell disease. This is a very timely and important topic, especially given the growing interest in non-drug approaches in SCD care. Authors clearly shows the clinical relevance of exercise and provide thoughtful insights into what helps or hinders its use. The qualitative approach is well chosen and seems to have been conducted carefully, with attention to COREQ guidelines and ethical considerations.

I do have a few suggestions that could make the manuscript even stronger:

• Add practical exercise examples: Including specific exercises suitable for children with SCD—like low-impact or graded activities—would give clinicians clearer takeaways and link perceptions to real-world practice.

• You discuss physiotherapists in the limitations, but acknowledging their role earlier could make the discussion of implementation barriers clearer.

• Check flow and readability: A few of the longer sentences could be tightened for smoother reading throughout the manuscript.

• Clean up references: There are a few repeated or inconsistently formatted references; standardizing them would improve overall presentation.

Overall, this is a strong and valuable manuscript, and addressing these points could make it even more reader-friendly and impactful.

**Do you want your identity to be public for this peer review?** For information about this choice, including consent withdrawal, please see our Privacy Policy

Reviewer #1: **Yes: ** Carlton Dampier MD

Reviewer #3: **Yes: ** Marta Wolosowicz

---

## [Author Response · Author response to Decision Letter 2]

6 Nov 2025

Response to Reviewer’s comments

Dear Editor,

We appreciate the reviewer’s valuable feedback on improving the clarity and specificity of the manuscript. Please find our detailed responses below for each comment.

Reviewer #3:

comment: Add practical exercise examples: Including specific exercises suitable for children with SCD—like low-impact or graded activities—would give clinicians clearer takeaways and link perceptions to real-world practice.

Response: We thank the reviewer for this important comment. However, we have added practical exercise examples. Highlighted on page 14 and read as “Such a program, like 'play-based & social activities' (indoor recreational games, light ball play, dancing, and performing daily chores under supervision),

Comment: You discuss physiotherapists in the limitations, but acknowledging their role earlier could make the discussion of implementation barriers clearer.

Response: We appreciate the reviewer’s insightful observation. The current study specifically explored healthcare professionals' (pediatric nurses’ and pediatricians’) perspectives on exercise as a therapy for sickle cell–associated musculoskeletal pain among children. As such, the inclusion of physiotherapists was beyond the original scope and participant design of the study. The intention was to capture how frontline medical and nursing professionals perceive, recommend, or integrate exercise within pediatric sickle cell care, given their frequent patient contact in clinical and ward settings. Participants acknowledged that their availability will enhance the implementation of the exercise in the management of MSK in SCD. However, we fully acknowledge that physiotherapists play an important role in exercise prescription, movement retraining, and graded activity planning for children with sickle cell disease. Their perspectives would indeed deepen understanding of implementation feasibility and enhance interdisciplinary insight. In line with this, we have clarified their potential contribution earlier in the discussion to strengthen the connection between professional collaboration and implementation barriers. We have also noted the need for future studies to include physiotherapists and other rehabilitation professionals to provide a more holistic view of exercise-based interventions for this population.

Comment: Check flow and readability: A few of the longer sentences could be tightened for smoother reading throughout the manuscript.

Response: Thank you for the observation. The longer sentences have been reviewed and revised throughout the manuscript to improve flow, clarity, and readability.

Comment: Clean up references: There are a few repeated or inconsistently formatted references; standardizing them would improve overall presentation.

Response: Thank you for the observation. The reference list has been thoroughly reviewed, and repeated or inconsistently formatted entries have been corrected.

---

## [Editor Report · Decision Letter 2]

11 Nov 2025

Exercise as a therapy for sickle cell associated musculoskeletal pain among children: Healthcare professionals’ perspectives.

PONE-D-25-29804R2

Dear Dr. Mensah Bonsu,

We’re pleased to inform you that your manuscript has been judged scientifically suitable for publication and will be formally accepted for publication once it meets all outstanding technical requirements.

Kind regards,

Tomasz W. Kaminski

Academic Editor

PLOS ONE

Additional Editor Comments: The Authors adequately addressed the minor comments raised by the Reviewer 3.

---

## [Editor Report · Acceptance letter]

PONE-D-25-29804R2

PLOS ONE

Dear Dr. Mensah Bonsu,

I'm pleased to inform you that your manuscript has been deemed suitable for publication in PLOS ONE. Congratulations! Your manuscript is now being handed over to our production team.

Kind regards,

on behalf of

Dr. Tomasz W. Kaminski

Academic Editor

PLOS ONE